

# The myogenic electric organ of *Sternopygus macrurus*: a non-contractile tissue with a skeletal muscle transcriptome

Matthew Pinch[1,*], Robert Güth[1,*], Manoj P. Samanta[2], Alexander Chaidez[1] and Graciela A. Unguez[1]

[1] Department of Biology, New Mexico State University, Las Cruces, NM, United States
[2] Systemix Institute, Redmond, WA, United States
[*] These authors contributed equally to this work.

## ABSTRACT

In most electric fish species, the electric organ (EO) derives from striated muscle cells that suppress many muscle properties. In the gymnotiform *Sternopygus macrurus*, mature electrocytes, the current-producing cells of the EO, do not contain sarcomeres, yet they continue to make some cytoskeletal and sarcomeric proteins and the muscle transcription factors (MTFs) that induce their expression. In order to more comprehensively examine the transcriptional regulation of genes associated with the formation and maintenance of the contractile sarcomere complex, results from expression analysis using qRT-PCR were informed by deep RNA sequencing of transcriptomes and miRNA compositions of muscle and EO tissues from adult *S. macrurus*. Our data show that: (1) components associated with the homeostasis of the sarcomere and sarcomere-sarcolemma linkage were transcribed in EO at levels similar to those in muscle; (2) MTF families associated with activation of the skeletal muscle program were not differentially expressed between these tissues; and (3) a set of microRNAs that are implicated in regulation of the muscle phenotype are enriched in EO. These data support the development of a unique and highly specialized non-contractile electrogenic cell that emerges from a striated phenotype and further differentiates with little modification in its transcript composition. This comprehensive analysis of parallel mRNA and miRNA profiles is not only a foundation for functional studies aimed at identifying mechanisms underlying the transcription-independent myogenic program in *S. macrurus* EO, but also has important implications to many vertebrate cell types that independently activate or suppress specific features of the skeletal muscle program.

Corresponding author
Graciela A. Unguez,
gunguez@nmsu.edu

## INTRODUCTION

Skeletal muscles are crucial for animals to perform behaviors that are necessary for survival in a changeable environment. These striated tissues are comprised of cells that,

although highly specialized for contraction and force generation, express a wide range of morphological, metabolic and structural properties. Characterizing different muscle cell phenotypes and determining how they are generated during development have been topics of intensive investigation (*Blais, 2005*; *Wigmore & Evans, 2002*; *Musumeci et al., 2015*). Since the phenotypic properties of fully differentiated muscle fibers are not fixed, numerous studies have also focused on identifying the processes involved in remodeling muscle fibers during postnatal stages (*Gundersen, 2011*; *Gunning & Hardeman, 1991*; *Schiaffino & Reggiani, 2011*). Cell signaling pathways and key regulatory factors associated with inducing not only the skeletal muscle phenotype, but specific muscle fiber type properties have also been discovered (*Buckingham & Rigby, 2014*; *Schiaffino & Reggiani, 2011*; *McCarthy, 2011*). Together, data from these studies have led to our current understanding that changes in environmental and intrinsic factors can affect gene transcription, or expression of genes, including turning genes completely on or off, or simply adjusting the levels of transcripts produced, and as a result influence muscle cell fate decisions throughout the life of the organism. Unfortunately, the existing evidence gives little insight into the coordination of subsets of muscle protein systems such as the genes associated with the different regions of the sarcomere, the structure that fully defines the adult skeletal muscle phenotype, and genes involved in sarcomere homeostasis. Studies on vertebrates wherein regulation of subsets of muscle genes are independently activated or repressed, and can be studied with high precision, are likely to further our understanding of some of these basic processes in muscle biology.

Electric fishes represent an excellent experimental model to address these unresolved questions. These vertebrate teleosts native to South America and Africa are unique in that they possess an electric organ (EO) that is specialized for the production of an electrical field outside the body (*Bennett, 1971*; *Bass, 1986*). This electric field is essential for navigation, communication and mate selection behaviors. In all electric fish species known to date, the current producing cells of the EO, i.e., electrocytes, derive from skeletal muscle cells (*Bennett, 1971*; *Fox & Richardson, 1978*; *Fox & Richardson, 1979*). Fully differentiated electrocytes, however, do not down-regulate the muscle program completely. In the gymnotiform *Sternopygus macrurus*, mature electrocytes are multinucleated target cells of cholinergic neurons like skeletal muscle fibers (*Unguez & Zakon, 1998*; *Cuellar, Kim & Unguez, 2006*). Electrocytes also contain muscle proteins like desmin, titin, $\alpha$-actin and $\alpha$-actinin (*Cuellar, Kim & Unguez, 2006*; *Kim et al., 2008*). However, unlike their precursor muscle cells, mature electrocytes do not contain sarcomeres or T-tubules (*Unguez & Zakon, 1998*). Interestingly, although fully differentiated electrocytes in *S. macrurus* transcribe some sarcomeric genes including slow and fast myosin heavy chain isoforms, and troponin-T, they do not contain the corresponding proteins (*Cuellar, Kim & Unguez, 2006*). Even more, members of the MyoD family of transcription factors (MTFs), i.e., *myoD*, *myogenin*, *myf5* and *myf6* (*mrf4, herculin*), are transcribed at levels that are similar to, or even higher, than those found in their skeletal muscle precursors (*Kim et al., 2004*; *Kim et al., 2008*). Moreover, these *S. macrurus*-specific MTFs can activate the myogenic program in mouse non-muscle cells following forced expression (*Kim et al., 2009*). That mature electrocytes in *S. macrurus* contain MTFs is intriguing given that expression of MTFs in other vertebrates

is linked to the activation of the striated muscle program (*Buckingham & Rigby, 2014*). Hence, unlike mammalian skeletal muscles, MTF-independent transcriptional events are likely to influence the homeostasis of muscle-specific properties of mature electrocytes in *S. macrurus* after differentiation raising an argument for determining the extent to which transcriptional regulation of myogenesis is conserved across different vertebrate species.

In the present study we extended our analyses of the regulation of sarcomere gene expression in *S. macrurus* electrocytes to include sets of genes associated with different regions of the sarcomere, i.e., the costamere, Z-disk and I-A-M band, protein degradation pathways including the ubiquitin-proteasome system (UPS) and the autophagy pathway, and transcription factors known to regulate the myogenic phenotype other than the MyoD family. In addition, we also performed deep RNA sequencing (RNA-seq) to more efficiently identify genes that are differentially expressed in muscle and electrocytes, and facilitate our investigations aimed at elucidating gene networks involved in modifying the myogenic program in electrocytes.

Our current data showed that all genes known to make up the mammalian sarcomere are transcribed in the mature EO of *S. macrurus*, and these were expressed at levels similar to those detected in skeletal muscle. Our expression analyses also established that genes belonging to protein degradation pathways in striated muscle were expressed very similarly in mRNA quantity in both muscle and EO tissues arguing against an elevated breakdown of sarcomeric proteins taking place in electrocytes. Moreover, our RNA-seq data revealed that the mature EO maintains a transcriptome profile that is over ninety percent comparable to that of its precursor striated muscle. Thus, mature electrocytes in *S. macrurus* represent myogenically derived cells that became highly specialized non-contractile electrogenic cells without changing their striated muscle transcriptome. Just as our transcript profile shows that virtually all transcripts associated with the sarcomere are similarly expressed in muscle and EO, we found that myomirs miR-1/133/206 are similarly expressed in both tissues. However, we identified a set of microRNAs associated with the regulation of contractile muscle phenotype including miR-30a/193b/365 that are enriched in EO. Our transcript expression analysis coupled with the differential expression of microRNAs described here provides new scenarios by which transcriptional and post-transcriptional events may regulate specific contractile properties in a vertebrate teleost.

## MATERIALS AND METHODS

*Sternopygus macrurus* were obtained commercially from Ornamental Fish (Miami, FL). Adult fish of undetermined sex and 30–50 cm in length were housed individually in 15- to 20-gallon aerated aquaria maintained at 25–28 °C and fed live red worms (Armstrong's Cricket Farm, West Monroe, LA) three times weekly. All fish used in this study had been in the aquaria for a minimum of one year. All fish were anesthetized in 2-phenoxyethanol (1.0 mL/L), and tail tissues were harvested for mRNA and microRNA (miRNA) detection and quantification using quantitative RT-PCR (qRT-PCR), deep RNA sequencing (RNA-seq), and *in situ hybridization*. In *S. macrurus*, there is a well-defined anatomical compartmentalization of skeletal muscle and EO tissues (*Unguez & Zakon, 1998*), which greatly

facilitates their dissection and isolation with little to no tissue cross-contamination. Ventral muscle and EO tissues were excised, immediately immersed in RNA*later*™ (Ambion, Austin, TX, USA) and stored at $-80\,^{\circ}$C until RNA extraction for subsequent mRNA analysis with RNA-seq ($n = 1$ fish), qRT-PCR ($n = 4$ fish), and miRNA analysis with qRT-PCR ($n = 3$ fish). In a fourth group of adult control fish ($n = 2$), the distal tail was amputated and frozen in isopentane cooled in liquid nitrogen for *in situ* hybridization processing. Following tissue and tail dissections, fish were euthanized by an overdose in 2-phenoxyethanol (1.0 mL/L). All animal experimental procedures strictly conformed to the American Physiological Society Animal Care Guidelines and were approved by the Institutional Animal Care and Use Committee at New Mexico State University (Approval # 2014-044).

## Determination of expression levels of mRNAs and miRNAs associated with sarcomeric structures in skeletal muscle and electric organ
### Approach: mRNA and miRNA analysis using qRT-PCR

Total RNA was extracted from muscle and EO of adult control fish using TRIzol® (Life Technologies). Concentration (ng/μL) and purity (260/280) measurements of total RNA from each muscle and EO sample were taken with a NanoDrop Lite spectrophotometer (Thermo Scientific). Total RNA was treated with DNase, and cDNA synthesis was performed as per manufacturer's instructions (SuperScript First-Strand Synthesis System kit; Invitrogen). To ensure the absence of genomic DNA in subsequent RNA analyses, we produced a set of RNA samples that were not reverse transcribed (no-RT samples) by substituting the reverse transcriptase enzyme with nuclease-free water. cDNA and no-RT samples were stored at $-20\,^{\circ}$C. Gene specific primers for mRNA and quantification (Table 1) were designed with Primer3Plus (*Koressaar & Remm, 2007*; *Untergasser et al., 2012*) and evaluated with the program NetPrimer (*Premier Biosoft, 2015*). All primer pairs were first tested with qualitative PCR using Taq polymerase before qRT-PCR was performed. To validate amplicon size and ensure the absence of non-specific amplicons, qualitative PCR products were visualized on agarose gels stained with SYBR® Safe (Invitrogen). Annealing temperatures and primer concentrations were optimized for each primer pair.

qRT-PCR was performed using Perfecta SYBR® Green Fastmix (Quanta Biosciences, Inc., Gaithersburg, MD) and PCR reactions were performed in triplicate (cDNA samples) or single (no-RT samples) in 96-well plates with clear caps on a DNA Engine Opticon® 2 system (Bio-Rad, Hercules, CA). PCR amplification was carried out using a profile that consisted of an initial 30s denaturation at 95 °C followed by 40 cycles of 1 s denaturation at 95 °C, 15s primer annealing at optimized temperature depending on the primers (Table 1), and 30s extension at 72 °C. For every PCR reaction, the fluorescence signal was measured at the end of the extension period. The PCR reaction was immediately followed by melting curve analysis from 65 °C to 90 °C in 0.2 °C steps with a holding time of 2s per step. Data was collected using the Opticon Monitor™ 3.1.32 software (Bio-Rad, Hercules, CA, USA).

To determine primer pair amplification efficiency, raw fluorescence data were analyzed using the real-time PCR Miner tool (*Zhao & Fernald, 2005*). Raw $C_t$ values for each

**Table 1  Oligonucleotide primers used for quantitative RT-PCR.** Sequences are shown for sense and antisense primers, and the annealing temperature ($T_m$) is given for each primer pair.

| Gene | Sense primer | Antisense primer | $T_m$ (°C) |
|------|--------------|------------------|-----------|
| rps11 | TACCCAGAATGAAAGGGCGTAT | CATGTTCTTGTGCCTCTTCTCG | 59 |
| snrpb | AAGATCAAGCCCAAAAATTCCA | CCTGAGGTGTCATTACCTGCTG | 59 |
| hsp70 | CGATTTCTACACCTCCATCACCA | ATGCTCTTGTTGAGTTCTTTGCC | 58 |
| hsp90a2 | AAGGACAACCAGAAGCACATCT | TCCTCTTCATCTTCAGGGAGC | 60 |
| plec | ATCCTCCAAATCCTCTGTTCGC | GATATGTCTGGTGGTCAATGAGTCC | 60 |
| ank2 | GATCTTCGCATGGCTATTGTG | TCTTCTCACCCACTTCTTCAGG | 60 |
| dmd | AGCACCTACTGATCCAGCACTACT | CTGCCTGTAGTTTCCTGTTCTCC | 58 |
| cryaba | CATCACCTCCTCACTCTCCTCG | TGACTTGTCCTCTTGAGTAATGGG | 58 |
| actn2 | GCTTGCTGCTGTGGTGCC | GTGTTCACAATGTCCTCGGC | 58 |
| myoz3 | AAGGATGTGATGATGGAGGAGC | TCTTTCAGAGGACCAGTGTACCC | 62 |
| myom1 | TTTCAAAGAAAGACGCAGGTGT | ACAACGGATGTGGAGAGGATGA | 57 |
| ckma | AACAGTCGTGGTGAGCGTAGG | GAACACTCCGCCAACAGAGG | 58 |
| ttnb | ATGTGCCTGAGGATGACGG | CAAACACTTCTGGTCTAGTTGGAG | 60 |
| myh10 | GAGCAGGAGGAGTACCAGCG | TGGAACTTGGTGTGTGAGCC | 58 |
| mylk4 | CGGACATCTCTGACGAGGC | GGAGCATGTGTGGAGTGGC | 62 |
| myh-f | AAAGGTACAAAGTCTTGAATGCCAGT | AGCCTGAGTCATAGTCACTAGATTTG | 60 |
| myh-s | AGAGAGCTTGAAAATGAGGTG | TTCATCGTGTACTTTCTTGGAA | 60 |
| myf5 | AACGCCATCCAGTACATTGAGAG | AATCTGAGACCAGACAGGACTGC | 60 |
| myod | ATATTCCGTTCAACATCACCTCT | AGAAGACACCTGCCTGCTTG | 58 |
| myog | CGCAGTGCCATACAGTACATCG | TCTGAGCCTGTGATGCTGTCTAC | 62 |
| myf6 | CCGAAACGGGATGTGACAG | CAGAGTGGCAGCCTTACGC | 60 |
| sox6 | TTCCACGGCAGCAAGAGC | CCTTCATCACCGCATCCTG | 60 |
| eya1 | CCACCTCTGTAGCAGACGGC | TGTACTGTGTCTGACCATACGCC | 60 |
| ube2a | CCCGTCCGAGAACAATATCA | CAGGGCTGTTAGGATTTGGC | 58 |
| psmb1 | TGGTGACTGTTTGACGCTGAC | CGCCTGCTTTGTAAGTGTCC | 58 |
| psmc2 | AAGAGGAGGAGAAAGACGACGG | GCTAAATCCCATAACGCAGGAG | 60 |
| psmc5 | GGTGTCCCTGATGATGGTGG | CAGACACTCGGATGAAGGTGC | 58 |
| psmc6 | ATTGACGGGCGTTTGAAGG | TCCAACAACATACCGAGGGC | 58 |
| becn1 | ACGTGGACAAGGGCAAGATC | CCCATTTGAGGTTGGTGAGC | 58 |
| atg12 | CTGTTACCGATACAGGAGACACTGG | AGAAAGCGGGAGATAAACTGGG | 60 |
| gabarapa | CGACGACAGATTTCCACGG | ATTGTTGACGAAGAAGAAGAGTGC | 60 |

transcript were uploaded into the RefFinder online tool (*Xie et al., 2012*) in order to identify appropriate reference genes for normalization. This tool uses four different programs for ranking potential internal reference genes—BestKeeper (*Pfaffl et al., 2004*), NormFinder (*Andersen, Jensen & Ørntoft, 2004*), geNorm (*Vandesompele et al., 2002*), and the Comparative Delta $C_t$ Method (*Silver et al., 2006*)—to determine a single ranking of the most appropriate genes to use as internal reference genes. RefFinder assumes a perfect primer pair amplification efficiency for each qRT-PCR reaction when ranking the expression stability of each transcript. However, to more accurately rank the most stably expressed genes, we had to consider the amplification efficiency for each primer pair (*De Spiegelaere et al., 2015*). We then re-analyzed our qRT-PCR data using BestKeeper,

**Table 2  TaqMan assay information used for miRNA quantitative RT-PCR.**

| Assay name | Context sequence |
| --- | --- |
| hsa-miR-206 | UGGAAUGUAAGGAAGUGUGUGG |
| hsa-miR-365 | UAAUGCCCCUAAAAAUCCUUAU |
| ssa-miR-193-3p | AACUGGCCCGCAAAGUCCCGCU |
| smac-miR-30a | UGUAAACAUCUUUGACUGAAAGC |
| ola-miR-30d | UGUAAACAUCCCCGACUGGAAGC |
| hsa-miR-1 | UGGAAUGUAAAGAAGUAUGUAU |
| hsa-miR-499 | UUAAGACUUGCAGUGAUGUUU |
| hsa-miR-205 | UCCUUCAUUCCACCGGAGUCUG |

NormFinder, and geNorm using the calculated primer pair amplification efficiencies for each gene. Results from the latter analyses confirmed the ranking of *rps11* (a 40S ribosomal subunit protein) and *snrpb* (a riboprotein component of the spliceosome) as the two most-stably expressed genes. $C_t$ values for all 29 transcripts associated with sarcomere homeostasis were imported into Microsoft Excel and analyzed using calculated primer pair efficiency adjustment and normalized against *rps11* and *snrpb* to derive fold expression differences between muscle and EO.

Quantification of miRNAs was performed using TaqMan® miRNA qPCR analysis (Applied Biosystems). Total RNA isolated from muscle and EO of adult control fish ($n = 3$) was reverse transcribed using miRNA-specific primers following the instructions outlined in the TaqMan® instruction manual. Primers for all qRT-PCR reactions for all miRNAs except for miR-30a were ordered from conserved pre-designed sets commercially available from TaqMan® (Table 2). The miR-30a reaction was specially designed by TaqMan® using an *S. macrurus*-specific miR-30a sequence (Table 2). All reverse transcription reactions were performed immediately prior to qPCR reactions.

Quantification of miRNA expression was performed using the reaction conditions specified by TaqMan® in 96-well plates with clear caps on a DNA Engine Opticon® 2 system (Bio-Rad, Hercules, CA, USA). All cDNA reactions were performed in triplicate, while no-RT reactions were performed on single samples. Data was collected using the Opticon Monitor™ 3.1.32 software (Bio-Rad, Hercules, CA, USA). Raw fluorescence data were analyzed using the real-time PCR miner tool (*Zhao & Fernald, 2005*) as described above. Based on its high miRNA expression stability according to the raw $C_t$ values using the RefFinder tool (*Xie et al., 2012*), miR-206 was used for miRNA normalization. $C_t$ values for all miRNAs were imported into Excel, analyzed as using calculated primer pair efficiencies, and normalized against miR-206 to derive fold-change differences between skeletal muscle and EO.

### Approach: mRNA and miRNA analysis using RNA-seq Illumina sequencing

*RNA preparation.* Skeletal muscle and EO tissues from one adult fish were blotted dry, weighed, and immediately frozen in liquid nitrogen. Tissues were then chopped and pulverized separately in liquid nitrogen using a mortar and pestle, re-suspended in TRIzol® reagent (Life Technologies) and their total RNA extracted following

**Table 3** Summary of transcriptome sequencing of muscle and electric organ of *Sternopygus macrurus*.

**(A) Library statistics**

| Illumina paired end library | Read pairs (each pair counted as 1) | Read size |
|---|---|---|
| 1. *S. macrurus* EO | 175,605,480 | 2 × 101 nt |
| 2. *S. macrurus* muscle | 171,612,181 | 2 × 101 nt |
| 3. 5-day Spinal Transection EO | 161,775,709 | 2 × 101 nt |
| 4. 5-day Spinal Transection muscle | 152,159,414 | 2 × 101 nt |
| 5. 2-day Spinal Transection EO | 158,340,025 | 2 × 101 nt |
| 6. 2-day Spinal Transection muscle | 157,943,982 | 2 × 101 nt |

**(B) Assembly statistics**

| Assembler | Assembled transcripts (#) | Mean size | N50 size |
|---|---|---|---|
| Trinity assembly of 1, 2 (from *Gallant et al., 2014*) | 326,623 transcripts | 1,287 bp | 3,045 bp |
| SOAPdenovo-trans of all six libraries | 642,455 transcripts (61,532 scaffolds; 580,923 additional contigs) | 2,273 bp—scaffolds, 468 bp—all | 1,358 bp |

manufacturer's instructions. Residual DNA was removed from total RNA samples by treating with DNase I, Amplification Grade (Invitrogen) prior to purification using phenol:chloroform:isoamyl alcohol extraction followed by isopropanol precipitation. cDNA libraries were constructed from purified RNA using the Illumina TruSeq RNA Sample Preparation (v.2) kit (San Diego, CA, USA). Libraries were sequenced using 100 bp paired-end reads (2 × 100 bp) on an Illumina HiSeq2000 at the Biotechnology Center of the University of Wisconsin in Madison, WI. An overview of the subsequent sequencing and assembly steps performed is outlined in Fig. 1.

*Transcriptome assembly.* An initial *de novo* assembly of 694,435,322 reads from EO and muscle libraries (Table 3A) was performed using the Trinity assembler (*Grabherr et al., 2011*) and first reported in *Gallant et al., (2014)*. Briefly, for that assembly, high quality reads were obtained by removing adaptor sequences, duplication sequences, ambiguous reads and low quality reads. Short read libraries from muscle and EO tissues were combined and quality control and filtering was performed using the FASTX-Toolkit (*Cold Spring Harbor Labs, 2010*) as well as scythe (*Buffalo, 2014*) and sickle (*Joshi & Fass, 2011*). Those reads were assembled using Trinity with default options. This assembly produced 326,623 sequence contigs representing 221,914 subcomponents. Of these sequences, 163,477 were at least 500 bp length and 63,408 were at least 2,000 bp in length. The average sequence length was 1,287 bp (Table 3B).

In this study, we performed an additional assembly of 1,954,873,582 reads from EO and muscle libraries plus four libraries taken from spinally transected fish. The four additional libraries used to assemble this new reference library consisted of skeletal muscle and EO tissues taken from spinally transected fish (R Güth et al., 2014, unpublished data). These four additional libraries were processed at the same time using the same methods as the control libraries we are describing. Illumina libraries were re-assembled using

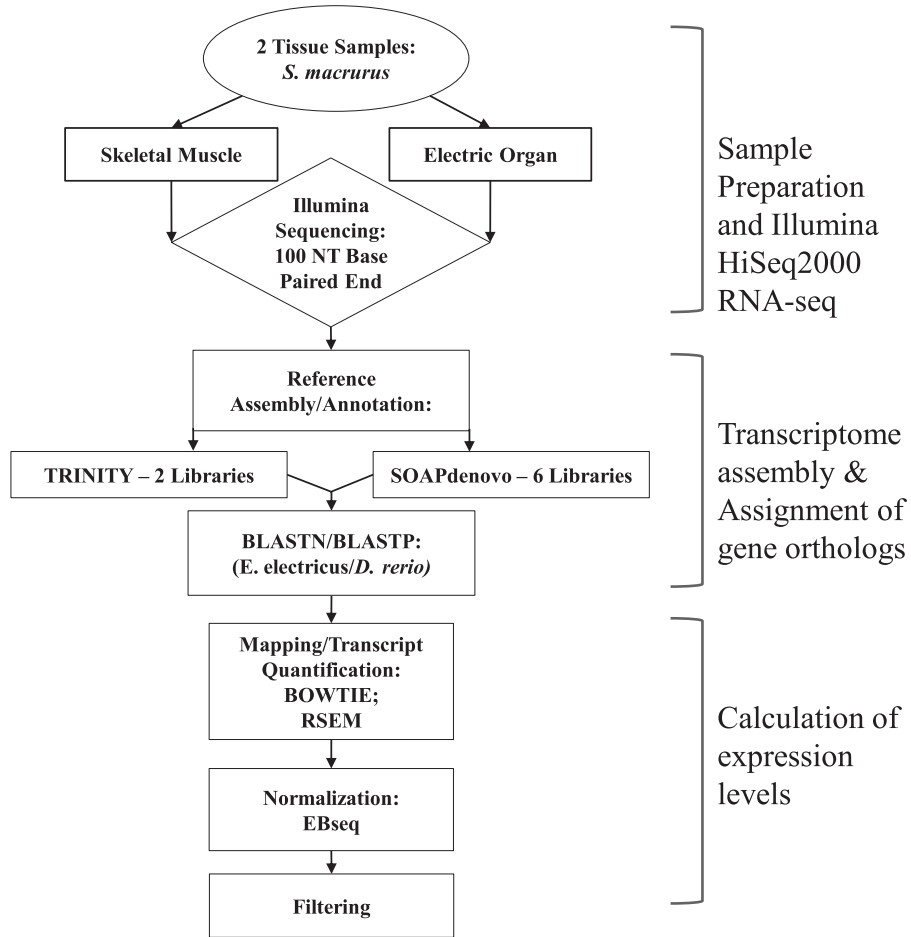

**Figure 1 Overview of steps performed to sequence, annotate and analyze transcriptomes of muscle and electric organ of *Sternopygus macrurus* using Illumina HiSeq2000.** Paired-end Illumina sequencing was performed on mRNA isolated from ventral muscle and electric organ of *S. macrurus* ($n = 1$). De novo reference transcriptomes were assembled and annotated using TRINITY (2 libraries) and SOAPdenovo (6 libraries), and confirmed by BLASTing against *D. rerio*. TRINITY and SOAPdenovo assemblies were combined to form one common reference transcriptome. Contigs were mapped to the reference using BOWTIE and quantified with RSEM. Muscle and electric organ transcript counts were normalized by library size using the Ebseq package in R. Finally, the annotated transcripts were filtered to eliminate any reads that contained less than 100 copies in both muscle and electric organ. Additional differential expression analysis using DESeq to validate our filtered results was also performed.

SOAPdenovo-trans (Table 3A), an assembler designed specifically for very large RNA-seq libraries with fast execution time and high optimization memory performance (*Xie et al., 2014*). The assembly was performed at kmer size $K = 31$. This assembly produced 61,532 scaffolds and 580,923 additional unassembled contigs. Of these sequences, 117,622 were at least 500 bp length and 28,883 were at least 2,000 bp in length. The average sequence length was 468 bp for all assembled transcripts and 2,273 bp for scaffolds only (Table 3B).

*Assignment of gene orthologs.* *S. macrurus* transcripts assembled from Trinity and SOAPdenovo-trans were compared separately against all genes from *Electrophorus electricus*

and *Danio rerio* using BLAST (blastn, *e*-value: 1e-5). Subsequently, all *S. macrurus* matches with each *E. electricus* gene were ranked from highest to lowest BLAST score. A similar but separate table was generated from comparisons between *S. macrurus* mRNA sequences and *D. rerio* genes. An *S. macrurus* transcript that matched one-to-one a pair of genes from *E. electricus* and *D. rerio* was selected as the *S. macrurus* ortholog for that gene. When more than one *S. macrurus* transcript matched a pair, the mRNA sequence with highest match based on CLUSTAL alignment with *E. electricus* and *D. rerio* was selected as the *S. macrurus* ortholog and used for analysis. In some cases when multiple *S. macrurus* transcripts had similar CLUSTAL-based sequence match to those of *E. electricus* and *D. rerio*, an average expression level was taken for each tissue type and an EO/muscle expression ratio was calculated from those average values. Manual confirmation of all *S. macrurus* transcripts discussed in this study was also performed.

*Calculation of expression levels.* Short reads from the muscle and EO libraries were individually mapped to the reference transcriptome assembly using bowtie (*Langmead et al., 2009*) (v.0.12.8) with default parameters. Read counting and ambiguity resolution was performed using RNA-Seq by Expectation Maximization (RSEM) (*Li & Dewey, 2011*) (v.1.2.3). Read counts were subsequently normalized for library size using the geometric mean method from EBSeq (*Leng et al., 2013*). The final data set comprised of 14,500 unique transcripts was generated after normalization and removal of sequences with fewer than 100 copies in both EO and skeletal muscle to minimize inclusion of genes likely to be expressed in cell types other muscle cells and electrocytes. A 4-fold difference in the expression of a transcript between muscle and EO was designated as being differentially expressed. Additionally, we quantified transcript numbers in each sample using Kallisto (*Bray et al., 2015*), a recently published efficient RSEM alternative, and computed differential gene expression between samples using DESeq2 (*Love, Huber & Anders, 2014*).

*High-throughput sequencing of miRNAs in muscle and EO of S. macrurus.* Total RNA for miRNA Illumina RNA-seq was isolated from the same muscle and EO samples that were used for the transcriptome sequencing described above ($n = 1$). The processing and analysis of the whole miRNA RNA-seq libraries has been described previously (*Gallant et al., 2014*; *Traeger et al., 2015*).

*Assignment of transcripts into functional pathways.* Of a total of 14,500 transcripts assembled, 12,508 were annotated with EntrezGene IDs for *D. rerio* orthologs. *D. rerio* EntrezGene IDs were obtained from the *D. rerio* genes database (Zv9) using the ID converter tool available at the BioMart Central Portal (*Smedley et al., 2015*). Differentially expressed genes were assigned color codes (EO > SM: red, EO < SM: blue, not differentially expressed: yellow) and *S. macrurus* ortholog EntrezGene IDs and corresponding color codes were submitted to KEGG Search&Color pathway analysis (*Kanehisa & Goto, 2000*) and searched against *D. rerio*.

## Statistical analysis of qRT-PCR studies

Normalized EO and muscle Ct values were grouped by tissue type (EO and muscle) and a paired student's $t$-test was used to determine which genes were differentially expressed. To minimize potential false-positives, $p$-values were adjusted using the Benjamini–Hochberg method in R (*R Core Team, 2016*). Samples with an adjusted $p$-value $< 0.05$ were considered to be differentially expressed. Transcript and miRNA qPCR data were analyzed separately.

## Prediction of transcript-miRNA interactions

3′ untranslated regions (3′ UTRs) from sarcomeric genes were isolated from their assembled transcript contigs and entered individually into the RNAhybrid web tool (*Rehmsmeier et al., 2004*) along with the *S. macrurus*-specific miRNA sequences for miR-365, miR-193b and miR-30a. Target predictions were performed using default alignment and energetic parameters with the exception of number of results to return, which were increased to 10 in order to ensure that all target sites for the same miRNA in each 3′ UTR would be identified if multiple target sites were present. In order to identify conservation of the predicted miRNA target sites in *S. macrurus* across other teleost species, miRNA target prediction was performed on 3′ UTRs and miRNAs of *D. rerio* using the TargetScan Fish release 6.2 web tool (*Lewis, Burge & Bartel, 2005*; *Ulitsky et al., 2012*).

## Sarcomeric mRNA expression in skeletal muscle and electric organ tissues using *in situ* hybridization

Serial tail cross-sections (16 μm thick) of control tails ($n = 2$) were cut in a cryostat and mounted on pre-chilled SuperFrost charged glass slides (VWR, Chester, PA). Prehybridization, hybridization, and washes were carried out per manufacturer's instructions (GeneDetect, Bradenton, FL). Briefly, tissue sections were fixed with 4% paraformaldehyde in 0.1 M PB (pH 7.4) at 4 °C for 15 min, washed three times for 5 min with 0.1 M PB, incubated with 100% ethanol for 5 min, and then air dried for 5 min at room temperature (RT). Oligonucleotide riboprobes (3′-DIG-labeled; GeneDetect) for fast myosin heavy chain (*myh-f*) and myosin light chain (*myl*) as well as for sense-strand negative control probes (Table 4) were diluted in hybridization buffer (20X SSC, 200 mg/ml dextran sulfate, 0.5X formamide, 10 mg/ml polyA, 10 mg/ml ssDNA, 10 mg/ml tRNA, 1 M DTT, 50X Denhardts) at 300 ng/ml and 350 ng/ml, respectively, slides were coverslipped and placed in a humidified hybridization chamber at optimized hybridization temperatures for each probe pair overnight for 18–21 hrs. All washes were performed at 55 °C unless otherwise noted and the hybridized probe was detected using the DIG Nucleic Acid Detection Kit (Roche, Germany). Tissue sections were blocked for 30 min in blocking solution (1:10 dilution; blocking reagent in maleic acid buffer (0.1 M maleic acid, 0.15 M NaCl, pH 7.5) at RT. The anti-digoxigenin-AP antibody was added (1:200 dilution) and slides were placed at 4 °C overnight. The slides were washed three times for 5 min with tris buffered saline (pH 7.5) and then equilibrated for 5 min with detection buffer (0.1 M Tris–HCl, 0.1 M NaCl, pH 9.5) at RT. The color substrate solution (200 μl of NBT/BCIP in 10 ml of detection buffer) was added and slides were placed in the dark overnight at RT. Images were captured with a Zeiss Axio Imager.Z1 epifluorescence microscope connected

**Table 4  Oligonucleotide probes used for *in situ* hybridization.**  Sequences are shown for antisense (a) and sense (s) negative control probes, and the hybridization temperature (°C). See text for abbreviations of genes.

| Gene | Probe | Sequence 5′–3′ | °C |
|---|---|---|---|
| myh-fast | a | CCTTTCCTGCTTCTCTGCTCTTGGCTCTGAGCTTGTTGACCTGGGACTCAGCGATGTCGGCAGTCCTGAGCTT CCTCCATCTCATGCTGCACCTTCCTGAACCTGGACAGGTGAGTGTTGGCCTGTCCTCAGCATCCTCAGCCTGT CTCTTGTAGGCTTTCACTTTCAGTTGCAGCTTGTCCACAAGACCTGCAGTCTTATCACATTCTTCTTGTCTTCCT CAGTCTGGTAGGTAAGCTCCTTCACTCTCCTTTCATATTTGCGCACTCCTTTAACAGCTTCAGCACTGCGTCTCT GCTCAGCTTCAACCTCACCTCCAGCTCACGCACCCTAGACTCCAGTTTCTGGAGCTGCT | 58 |
|  | s | GAAAGAAGCAGCTCCAGAAACTGGAGTCTAGGGTGCGTGAGCTGGAGGGTGAGGTTGAAGCTGAGCAGAGAC GCAGTGCTGAAGCTGTTAAAGGAGTGCGCAAATATGAAAGGAGAGTGAAGGAGCTTACCTACCAGACTGAGG AAGACAAGAAGAATGTGATAAGACTGCAGGATCTTGTGGACAAGCTGCAACTGAAAGTGAAAGCCTACAAGAG ACAGGCTGAGGATGCTGAGGAGCAGGCCAACACTCACCTGTCCAGGTTCAGGAAGGTGCAGCATGAGATGG AGGAAGCTCAGGAGCGTGCCGACATCGCTGAGTCCCAGGTCAACAAGCTCAGAGCCAAGAGCAGAGAAGCA GGAAAGGGAAGCGGCC | 58 |
| myl | a | AGCACCTTGAAAGAGGACAGGATGACATCCTCAGGGTCAGCACCCTTCAGCTTCTCTCCGAACATGGTGAGG AAGACGGTGAAGTTGATGGGGCCAGGGGCCTCCTTGATCATGGCCTCCAGCTCCTCATTCTTCACATTCAGCT GGCCCATAGAGGCCAGCACGTCCCTAAGGTCGTCCTTGCTGATGATACCATCTCTGTTCTGGTCAATGATTGT GAAAGCCTCTTTGTCTCCTGAA | 58 |
|  | s | CCAGATTCAGGAGACAAAGAGGCTTTCACAATCATTGACCAGAACAGAGATGGTATCATCAGCAAGGACGACC TTAGGGACGTGCTGGCCTCTATGGGCCAGCTGAATGTGAAGAATGAGGAGCTGGAGGCCATGATCAAGGAGG CCCCTGGCCCCATCAACTTCACCGTCTTCCTCACCATGTTCGGAGAGAAGCTGAAGGGTGCTGACCCTGAGGAT GTCATCCTGTCCTCTTTCAAGGTGCTGGATCCA | 58 |

to a Zeiss Axiocam ICc1 camera interfaced to a PC running Zeiss Axiovision Release 4.8.2 SP3 imaging software.

# RESULTS

## Mature electric organ of *S. macrurus* transcribes the full complement of sarcomere genes

In this study, transcript levels of genes associated with different regions of the sarcomere were characterized in skeletal muscle and EO of adult *S. macrurus*. Specifically, mRNA levels of 13 genes representing the costamere (*ankyrin*(*ank2*), *alpha-crystallin* (*cryaba*), *dystrophin* (*dmd*), and *plectin* (*plec*)); Z-disk (*α-actinin-2* (*actn2*), *myozenin* (*myoz3*)); and I-A-M band (muscle creatine kinase (*ckma*), *myosin heavy chain 10*(*myh10*), *fast myosin heavy chain* (*myh-f*), *slow myosin heavy chain* (*myh-s*), *myosin light chain kinase 4* (*mylk4*), *myomesin* (*myom1*), and *titin* (*ttnb*)) regions were quantified using qRT-PCR. Our data show that of these 13 genes, only the transcript levels for the costamere-associated *ank2* gene were significantly up-regulated in EO relative to muscle (Fig. 2A and Table 5A). Although mRNA levels for both *myh-f* and *myh-s* were more than 4-fold lower in EO relative to muscle, neither of these transcripts were significantly down-regulated (*myh-f*: $p = 0.25$; *myh-s*: $p = 0.09$) (Fig. 2A and Table 5A). Detection of similar *myh-f* and *myosin light chain* (*mlc*) mRNA expression in skeletal muscle and EO tissues by qRT-PCR was confirmed by *in situ hybridization* (Fig. 3). These data also demonstrated that detection of sarcomeric transcripts in adult EO was not due to contamination of the EO with skeletal muscle during the tissue dissections. Transcripts for genes known to play a role in mediating sarcomere development among other cellular functions, i.e., *heat shock cognate*

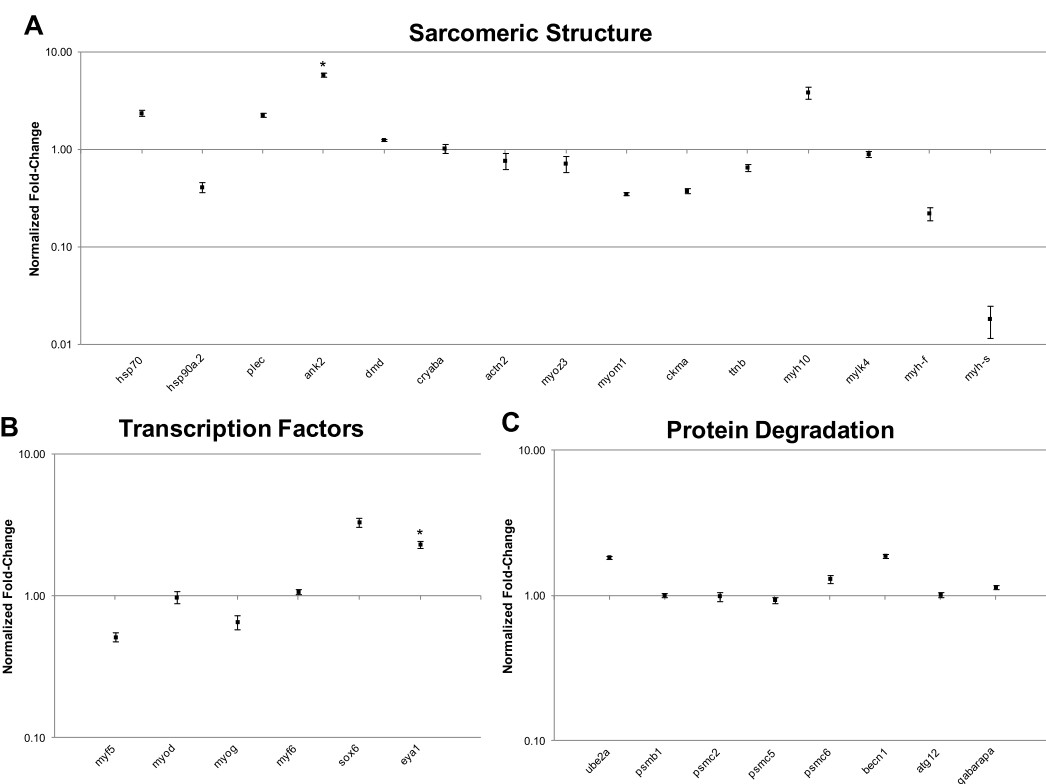

**Figure 2** **qRT-PCR analysis of 29 transcripts associated with the expression of sarcomeric genes.** Plots illustrating fold-change differences between EO and muscle in *S. macrurus* (*n* = 4). (A) Fold-change differences of 15 transcripts associated with sarcomere formation and structure. (B) Fold-change differences of six transcription factors associated with regulating the expression of sarcomeric genes. (C) Fold-change differences of eight genes involved in protein degradation pathways. Data are shown as fold-changes ± standard error, graphed on a logarithmic scale with values >1 indicating increased expression in EO relative to muscle and values <1 indicating decreased expression in EO relative to muscle. Data points marked with asterisks are significantly differentially expressed between EO and muscle (adjusted *p*-value < 0.05).

*70-kd protein* (*hsp70*) and *heat shock protein 90, alpha (cytosolic), class A member 1, tandem duplicate 2 hsp90aa1.2*)), *Hellerschmied & Clausen, (2014)* were also detected in muscle and EO at similar levels (Fig. 2A and Table 5A).

The Illumina high-throughput sequencing of transcriptomes from the skeletal muscle and EO of one adult fish provided us with a complete profile of the sarcomere genes expressed in these myogenic tissues beyond those studied using qRT-PCR. These data showed that all genes known to make up the mammalian sarcomere (*Ehler & Gautel, 2008*; *Treves et al., 2009*) are transcribed in the fully differentiated EO of *S. macrurus*. Specifically, we identified 112 mammalian orthologs that compose the I-A-M band (Fig. 4A), Z-disk region (Fig. 4B), and costamere (Fig. 4C). We also found the EO to contain 96 of 112 sarcomeric gene transcripts at levels that were similar to those found in muscle (Fig. 4). At least eight different sarcomeric myosin heavy chain (MHC) mRNAs corresponding to different muscle cell types were detected and since their expression levels did not differ between tissue types, we represented this similarity in MHC isoform mRNAs and quantity

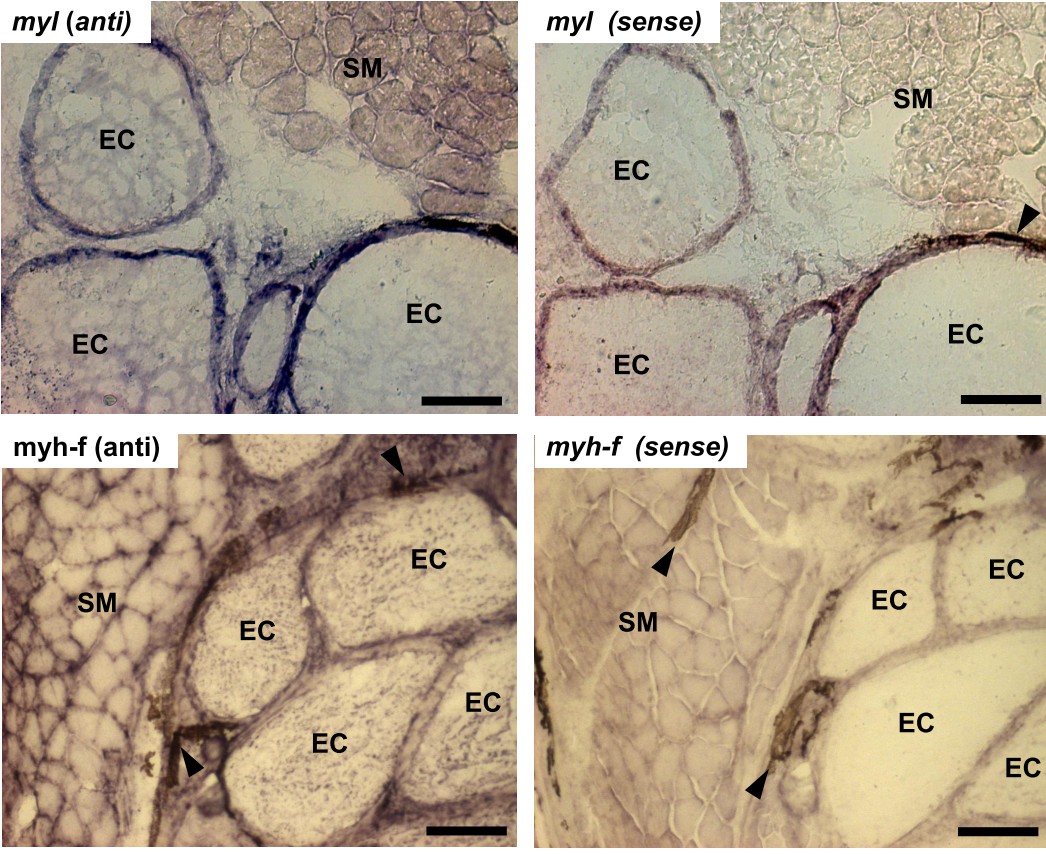

**Figure 3** *In situ* **hybridization of muscle and electrocytes using probes against myosin light chain and fast-myosin heavy chain.** Portions of transverse cryosections (16 μm-thick) taken from tails of control fish and processed for *in situ* hybridization. Antisense (anti) DIG-labeled RNA riboprobes specific to *S. macrurus* myosin light chain (*myl*) and sarcomeric fast myosin heavy chain (*myh-f*) transcripts were used for detection of these mRNAs in skeletal muscle fibers and electrocytes (EC). Sense strand DIG-labeled riboprobes against myosin light chain (*myl*) and sarcomeric fast myosin heavy chain (*myh-f*) were used as negative controls. Melanin is the dark brown pigment found around muscle fibers and electrocytes (arrow heads). Abbreviations: SM, skeletal muscle fibers; EC, electrocyte. Scale bar, 100 μm.

by providing only one general category for sarcomeric "myosin heavy chain" in Fig. 4. Of the 16 genes that were not similarly expressed in muscle and EO, three were detected in the EO at levels lower than 4-fold compared to muscle—the glycolytic enzyme *beta-enolase* (*eno3*) (EO: 22,939; muscle: 166,518 copies), *actn2* (EO: 6,055; muscle: 30,011 copies) and *atp1a1a.,2* (EO: 311; muscle: 3,235 copies). *eno3* is localized to the M-band and predominant in fast-twitch fibers (*Foucault et al., 1999*) (Fig. 4A). The thirteen genes that were more than 4-fold upregulated in EO are associated with the actin cytoskeleton that promotes cell attachment and these include: *desmin* (*desmb*), *ank2*, *cryaba*, *N-cadherin* (*cdh4, cdh8, cdh12*), and *alpha-actinin-2-associated LIM protein* (ALP; *pdlim3a*) (*Clark et al., 2002*) (Fig. 4C). DEseq analysis supported the 4-fold cut-off we established, as all three down-regulated genes were called by DESeq too, and only three up-regulated genes *ank2* ($p = 0.12$), *cdh4* ($p = 0.07$), and *cryaba* ($p = 0.07$) were not called as differentially expressed ($p \leq 0.05$) by DESeq.

Pinch et al. (2016), *PeerJ*, DOI 10.7717/peerj.1828

**Table 5  Comparison of fold change differences of 29 transcripts between EO and muscle measured by qRT-PCR and RNA-seq.** qRT-PCR measurements were normalized against the geometric means of two internal reference genes: rps11, and snrpb. For all qRT-PCR ratios, $n = 4$. Statistical analysis of the qRT-PCR dataset identified five significantly differentially expressed transcripts ($p \leq 0.05$, designated '*'). All RNA-seq ratios ($n = 1$) were considered to be differentially expressed if the difference in transcript expression was more than 4-fold different between EO and muscle. †—To date, we have not resolved the identity of all fast-twitch and slow-twitch myosin heavy chain isoforms transcribed in muscle and EO of *S. macrurus* and therefore we did not include RNAseq data for *myh-f* and *myh-s* in this table.

**(A) Sarcomeric structure**

| | | *hsp70* | *hsp90a2* | *plec* | *ank2* | *dmd* | *cryaba* | *actn2* | *myoz3* | *myom1* | *ckma* | *ttnB* | *myh10* | *mylk4* | *myh-f* | *myh-s* |
|---|---|---|---|---|---|---|---|---|---|---|---|---|---|---|---|---|
| qRT-PCR | Fold-change | 2.37 | 0.41 | 2.28 | 5.89* | 1.26 | 1.03 | 0.77 | 0.72 | 0.35 | 0.38 | 0.66 | 3.88 | 0.90 | 0.22 | 0.02 |
| | Standard error | 0.16 | 0.05 | 0.10 | 0.34 | 0.02 | 0.12 | 0.14 | 0.13 | 0.01 | 0.02 | 0.05 | 0.57 | 0.07 | 0.03 | 0.01 |
| | *P*-value | 0.08 | 0.18 | 0.10 | 0.04 | 0.38 | 0.99 | 0.65 | 0.38 | 0.08 | 0.14 | 0.66 | 0.08 | 0.99 | 0.26 | 0.09 |
| RNA-seq | Fold-change | 3.54 | 0.70 | 3.61 | 4.75 | 2.24 | 6.46 | 0.20 | 1.50 | 0.62 | 0.70 | 0.70 | 12.19 | 0.33 | † | † |

**(B) MTFs**

| | | *myf5* | *myoD* | *myog* | *myf6* | *sox6* | *eya1* |
|---|---|---|---|---|---|---|---|
| qRT-PCR | Fold-change | 0.51 | 0.97 | 0.65 | 1.07 | 3.30 | 2.30* |
| | Standard error | 0.04 | 0.10 | 0.07 | 0.05 | 0.24 | 0.12 |
| | *P*-value | 0.18 | 0.99 | 0.08 | 0.96 | 0.11 | 0.05 |
| RNA-seq | Fold-change | 0.67 | 1.68 | 1.21 | 2.39 | 5.54 | 6.81 |

**(C) Protein degradation**

| | | *ube2a* | *psmb1* | *psmc2* | *psmc5* | *psmc6* | *becn1* | *atg12* | *gabarapa* |
|---|---|---|---|---|---|---|---|---|---|
| qRT-PCR | Fold-change | 1.82 | 1.00 | 0.98 | 0.92 | 1.29 | 1.86 | 1.00 | 1.13 |
| | Standard error | 0.05 | 0.03 | 0.07 | 0.05 | 0.09 | 0.07 | 0.04 | 0.03 |
| | *P*-value | 0.12 | 0.99 | 0.99 | 0.72 | 0.38 | 0.10 | 0.99 | 0.64 |
| RNA-seq | Fold-change | 2.35 | 1.40 | 1.66 | 1.69 | 1.84 | 2.89 | 2.34 | 2.22 |

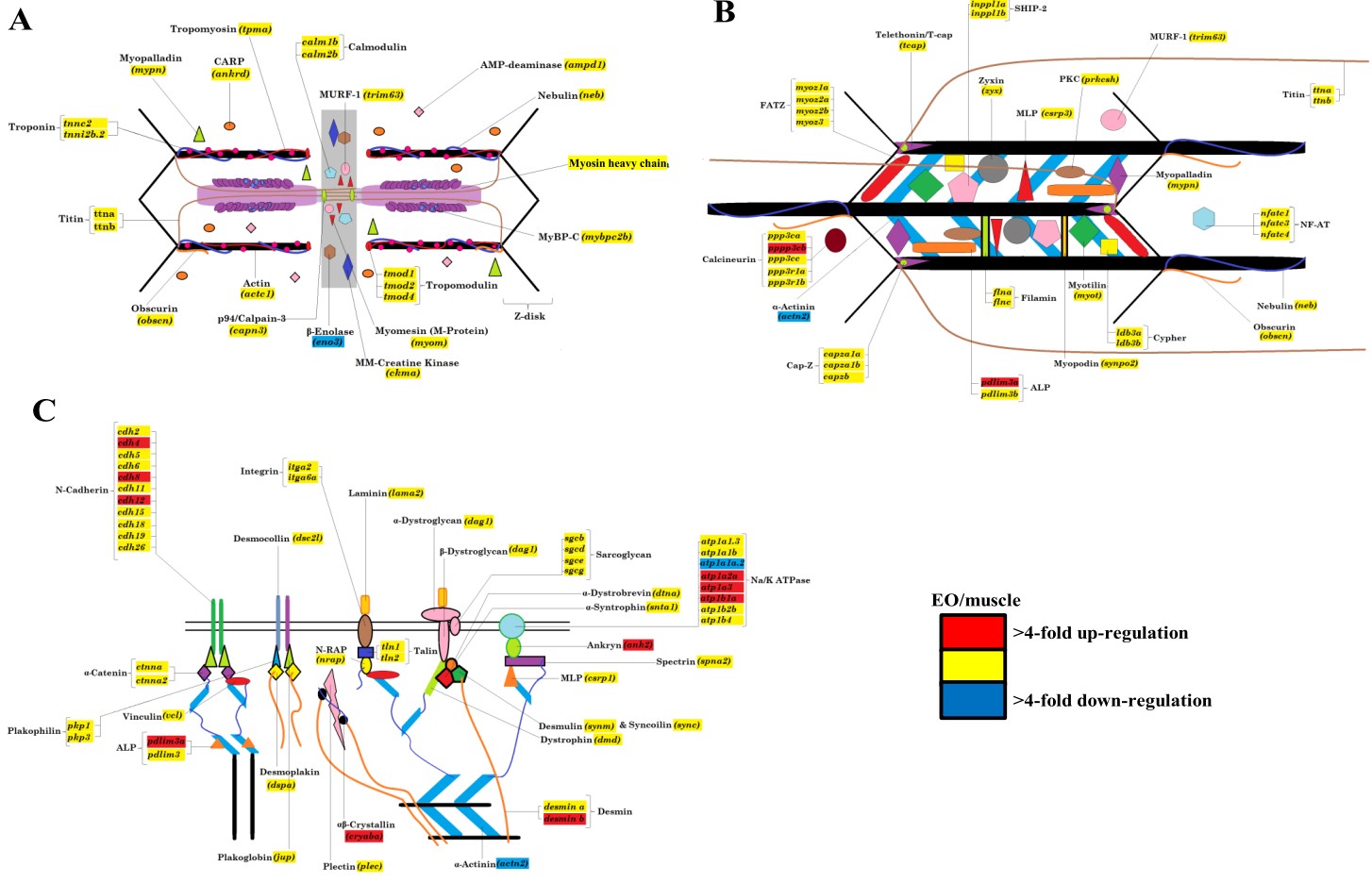

**Figure 4** **Model of the contractile apparatus and the relative expression levels of transcripts representing sarcomeric components in muscle and EO.** (A) I-band, A-band, and M-line regions of the sarcomere; (B) sarcomere Z-disk region; and (C) cytoskeletal filament linkages at the sarcolemma of striated muscle. All sarcomeric genes correspond to those listed for mammalian sarcomeres in *Ehler & Gautel (2008)*. Transcripts labeled in red, yellow and blue represent transcripts expressed at levels >4-fold higher in EO than muscle, similar in both EO and muscle, and <4-fold lower in EO than muscle, respectively. To date, we have not resolved the identity of all myosin heavy chain isoforms transcribed in muscle and EO of *S. macrurus* and thus, all identified isoforms have been included in one category, labeled "Myosin Heavy Chain" in this figure.

## Myogenic transcription factors are not downregulated in the mature electric organ of *S. macrurus*

Previous expression studies in *S. macrurus* showed that the adult EO continues to transcribe the myogenic transcription factors (MTFs) *myf5*, *myod*, *myogenin* (*myog*) and *myf6* (*mrf4*) (*Kim et al., 2004*; *Kim et al., 2008*) and moreover, these *S. macrurus*-specific MTFs are capable of inducing the myogenic phenotype in mammalian non-muscle cells *in vitro* (*Kim et al., 2009*). In this study, we expanded our expression analysis of MYFs with important roles in myogenesis. Specifically, the expression levels of *srf*, *prdm1*, *six1*, *sox6*, *eya1*, *fox03*, *hdac*, and members of the MEF2 (*mef2ca, mef2cb, mef2d*) and NFAT (*nfatc1, nfatc3, nfatc4*) families of transcription factors were studied using qRT-PCR and/or RNA-seq (Figs. 2B, 6 and Table 5B). In general, our data showed that all transcription factors are transcribed in adult EO tissue at levels similar (<4-fold difference) to those found in skeletal muscle. Only

*sox6* and *eya1* were upregulated (<4-fold difference) in EO relative to skeletal muscle in the RNA-seq study, and only *eya1* was significantly upregulated ($p < 0.05$) in our qRT-PCR study (Figs. 2B, 6 and Table 5B). In addition, our RNA-seq and qRT-PCR data corroborated our previous findings showing that the four members of the MyoD family of MTFs are not down-regulated in mature EO tissue despite its lack of sarcomeric structures. DESeq corroborated results that we observed in our qRT-PCR analysis, and the 4-fold cut-off that we imposed on our transcriptome data by calling all six transcription factors analyzed by qRT-PCR as similarly expressed. This corroborates our qRT-PCR data in five of six cases with eya1 as the only exception, being called significantly upregulated by qRT-PCR ($p = 0.05$), and not significantly different by DESeq ($p = 0.07$).

### Components of protein degradation pathways are not differentially expressed in muscle and EO of *S. macrurus*

Since our qRT-PCR and RNA-seq data showed that all sarcomere components and key myogenic transcription factors that regulate many of these contractile genes are transcribed in adult EO at similar, or even higher levels than those found in skeletal muscle, we examined some of the key genes associated with protein degradation pathways including the ubiquitin-proteasome system (UPS) and the autophagy pathway (Fig. 2C and Table 5C). Based on qRT-PCR, expression levels of all UPS genes (*ube2a*, *psmb1*, *psmc2*, *psmc5*, *psmc6*) were not significantly different between muscle and EO tissues (Fig. 2C and Table 5C). Similarly, genes in the autophagy pathway such as *becn1*, *atg 12*, and *gabarapa* were not significantly differentially expressed between muscle and EO (Fig. 2C and Table 5C). The expression levels detected by qRT-PCR were corroborated by the RNA-seq survey (Table 5C). In addition to all genes in both the UPS and autophagy pathways (Figs. S1–S3), the RNA-seq survey also identified other known genes reported to be associated with protein turn-over including *capn3*, *trim63* (MuRF1) and *fbxo32* (Atrogin-1) (*Duguez, Bartoli & Richard, 2006*; *Witt et al., 2008*; *Bonaldo & Sandri, 2013*)—all transcripts with expression levels similar between EO and skeletal muscle (Fig. 6). Hence, based on the mRNA expression of components belonging to protein degradation pathways, these data imply that post-translational regulation of sarcomeric proteins might not play a major role in maintaining a partial muscle phenotype in the adult EO of *S. macrurus*. DESeq analysis supported the similar expression of proteolytic genes that we observed, with all eight genes detected in qRT-PCR also being called not significant ($p > 0.05$) in DESeq.

### A set of microRNAs are predicted to regulate sarcomeric components in the EO of *S. macrurus*

The absence of any indication that mature electrocytes downregulate the expression of genes that code for sarcomere structures at the mRNA level while retaining the expression of only a subset of its contractile proteins is suggestive of the involvement of post-transcriptional events in regulating the skeletal muscle program in the EO. This led us to perform an Illumina sequencing of mature miRNAs isolated from muscle and EO of *S. macrurus*. Our miRNA sequencing and expression analysis revealed 155 conserved miRNAs with known functions, all of which were expressed in both EO and muscle. Four miRNAs associated with the skeletal muscle phenotype were detected in EO: miR-133, miR-1, miR-206, and

**Table 6   Comparison of select miRNA fold-changes between EO and muscle across both TaqMan® qRT-PCR and Illumina RNA-seq platforms.** TaqMan® qRT-PCR measurements were normalized against miR-206. For all qRT-PCR ratios, $n = 3$. Statistical analysis of the qRT-PCR dataset identified one significantly differentially expressed miRNA ($p \leq 0.05$; designated '*'). All RNA-seq ratios ($n = 1$) were considered to be differentially expressed if the difference in miRNA expression was more than 4-fold different between EO and muscle.

|  |  | miR-365 | miR-193b | miR-30a | miR-30d | miR-1 | miR-499 | miR-205 |
|---|---|---|---|---|---|---|---|---|
|  | Fold-change | 9.70* | 7.40 | 4.31 | 2.35 | 0.99 | 0.58 | 0.26 |
| qRT-PCR | Standard error | 0.31 | 0.53 | 0.37 | 0.15 | 0.09 | 0.02 | 0.006 |
|  | *P*-value | 0.04 | 0.06 | 0.06 | 0.06 | 0.92 | 0.07 | 0.16 |
| RNA-seq | Fold-change | 14.79 | 8.88 | 21.82 | 1.32 | 1.52 | 0.49 | 0.09 |

miR-499. Their expression levels were similar in both muscle and EO tissues (Table 6). Three miRNAs that were upregulated in EO relative to muscle included miR-365 (~15 fold), miR-193b (~9 fold), and miR-30a (~22 fold) (Table 6). miR-365 and miR-193b have been implicated in the loss of sarcomeric gene expression during their induction of adipogenesis in C2C12 cells (*Sun et al., 2011*), while miR-30a is involved in early muscle fate-determination (*Soleimani et al., 2012*; *Ketley et al., 2013*; *Guess et al., 2015*).

We confirmed the RNA-seq miRNA profiles using TaqMan® qRT-PCR for a total of eight miRNAs including the three highly upregulated miRNAs (miR-365, miR-193b, miR-30a), three muscle-specific miRNAs or myomirs (miR-1, miR-206, miR-499), one that was similarly expressed in both muscle and EO tissues (miR-30d), and one that was highly downregulated in EO (miR-205). In general, the relative trends and differences in miRNA expression detected by RNA-seq were confirmed by our qRT-PCR data (Fig. 5 and Table 6). In EO, miR-365 (9.7-fold; $p = 0.04$) and miR-193b (7.4-fold; $p = 0.06$) were highly upregulated while the three myomirs and miR-30d were detected in similar amounts in both EO and skeletal muscle and miR-205 levels were lower in EO than in muscle (Fig. 5 and Table 6). Discrepancies obtained between RNA-seq and qRT-PCR methodologies were related to the magnitude, but not direction, of the expression levels detected in muscle and EO. For example, miR-30a was upregulated about 22-fold in EO according to RNA-seq, but only upregulated about 4.3-fold based on qRT-PCR (Fig. 5 and Table 6). It was not unexpected to observe some variation in the data using these two detection platforms given the biological differences between fish sampled and a sample size of one used for the RNA-seq miRNA survey.

Following confirmation of the expression levels of miRNAs upregulated in EO that play roles in regulating the muscle phenotype, we determined whether they also contained sequences complementary to the 3′ UTRs of sarcomeric transcripts, which might suggest their potential to regulate their translation. Based on the analysis using *S. macrurus*-specific 3′ UTR sequences, miRNA sequences on the RNAhybrid miRNA target prediction program, *D. rerio* 3′ UTR sequences, and miRNA sequences on the TargetScan Fish release 6.2 program, we found that the highly upregulated miR-30, miR-365, and miR-193b in mature EO had conserved target sites in several genes associated with the sarcomere: *calmodulin 2b* (miR-30 and miR-365), I-A-M band-associated *tropomodulin 1* (miR-193b), and M-line associated *myomesin* (miR-30) (Fig. 6).

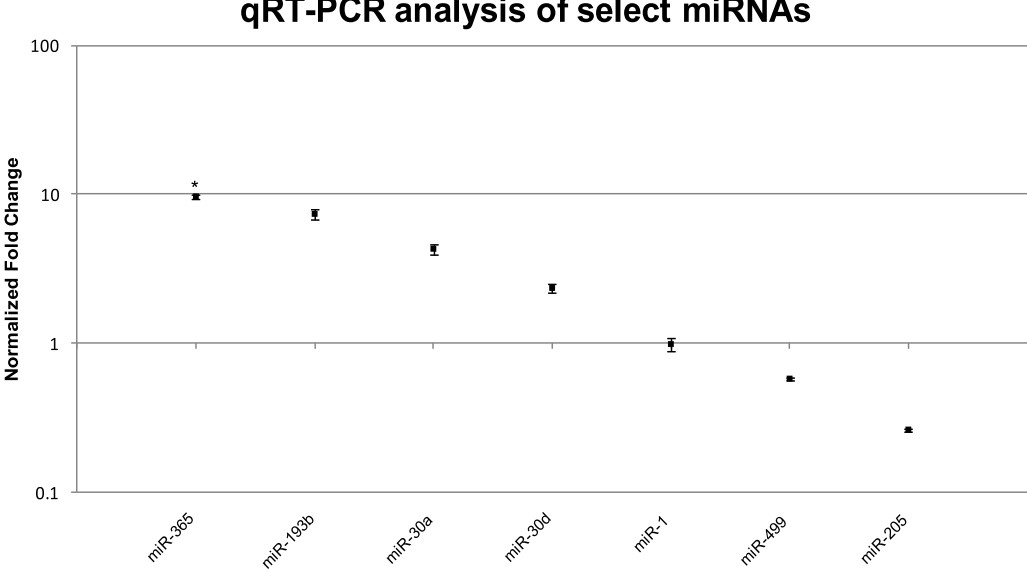

**Figure 5** **TaqMan® qRT-PCR analysis of seven miRNAs in EO and muscle of *S. macrurus*.** Plots illustrating fold-change differences between EO and muscle in *S. macrurus* ($n = 3$). Data are shown as fold-changes ± standard error, graphed on a logarithmic scale with values >1 indicating increased expression in EO relative to muscle and values <1 indicating decreased expression in EO relative to muscle. Data points marked with asterisks are significantly differentially expressed between EO and muscle (adjusted *p*-value < 0.05).

## DISCUSSION

The analyses presented here describe the molecular expression of a subset of contractile genes in the mature EO and skeletal muscle tissues of *S. macrurus*. These data expand previous findings by increasing our transcript characterization of genes associated with sarcomere homeostasis using qRT-PCR. Our deep RNA sequencing of transcriptomes and miRNA compositions of muscle and EO tissues from an adult fish informed our qRT-PCR analysis on expression of sarcomere-associated genes, allowed the identification of additional genes that might be differentially expressed in muscle and electrocytes, and facilitated our investigations aimed at elucidating gene networks involved in modifying the myogenic program in electrocytes.

Our current transcript analysis revealed that electrocytes and muscle fibers express all components that make up a sarcomere in mammalian striated muscle fibers (*Ehler & Gautel, 2008*; *Treves et al., 2009*). Most unexpected was the finding that electrocytes retained the expression of all these genes at levels similar to, or even higher than, those detected in myofibers. Examination of transcription factors known to regulate sarcomere gene expression also showed no differences, in either quality or quantity, between muscle and EO (Fig. 6). Given the role of degradation pathways in the homeostasis of molecular composition, we then predicted genes associated with these pathways to be significantly upregulated in EO, but were unable to find changes in the expression levels of key subsets of genes associated with the ubiquitin-proteasome system (UPS) and the autophagy pathway (Fig. 6; Figs. S1–S3). These data do not support protein degradation as a key mechanism by

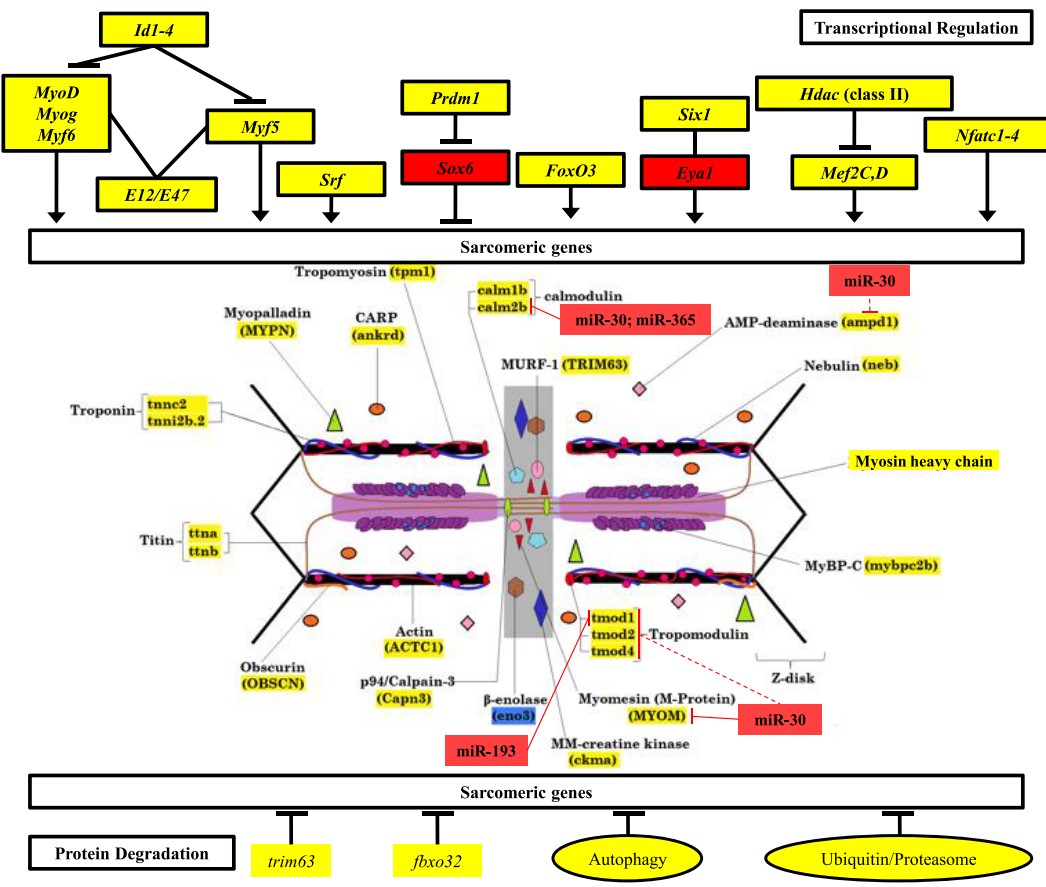

**Figure 6** **Mechanisms of control of sarcomeric gene expression in skeletal muscle and electric organ of** *S. macrurus.* Our qRT-PCR and transcriptome analysis of the expression of sarcomeric genes in muscle and EO of *S. macrurus* demonstrates that all transcription factors associated with maintenance of sarcomeric gene expression are similarly expressed, or even upregulated in the EO of *S. macrurus* relative to muscle (Top). Concomitantly, virtually all sarcomeric genes are transcribed at similar levels in both EO and muscle (middle), suggesting that transcriptional regulation is not the main driver of the loss of sarcomeres in EO. Additionally, genes associated with protein degradation are similarly expressed in EO and muscle (bottom), suggesting that protein degradation should not differ between both tissues. Finally, we identified a set of upregulated miRNAs and predicted a set of targets within the sarcomere (middle) that provide evidence of an important role of post-transcriptional regulation in controlling the partial muscle phenotype observed in the EO of *S. macrurus*. Names highlighted in red represent transcripts that were >4-fold upregulated in EO relative to muscle; Names highlighted in yellow represent transcripts that are similarly expressed in EO and muscle; Names highlighted in blue represent transcripts that were >4-fold upregulated in muscle relative to EO. miRNA target predictions with dashed lines represent targets predicted in *D. rerio* only using TargetScan Fish v. 6.2 (*Lewis, Burge & Bartel, 2005*; *Ulitsky et al., 2012*); miRNA target predictions with solid lines represent conserved targets in both *D. rerio* and *S. macrurus* as predicted by TargetScan Fish v. 6.2 (*Lewis, Burge & Bartel, 2005*; *Ulitsky et al., 2012*) and RNAhybrid (*Rehmsmeier et al., 2004*; *Krüger & Rehmsmeier, 2006*).

which mature electrocytes selectively repress various components of the contractile muscle program. Of interest, analysis of the EO transcriptome revealed that only 2% of the 14,500 transcripts comprising our reference transcriptome generated from muscle and EO tissues were downregulated, and 8% were upregulated in mature electrocytes in comparison to muscle fibers. Upon examination of this set of differentially expressed genes, we were

unable to identify genes with a known link to the synthesis, maintenance or degradation of the sarcomere complex. These data (Fig. 6; Figs. S1–S3) combined with previous studies demonstrating partial expression of sarcomeric proteins in *S. macrurus* EO (*Unguez & Zakon, 1998*; *Cuellar, Kim & Unguez, 2006*), can most parsimoniously be interpreted to mean that the transcription of a striated muscle program in the absence of a contractile phenotype requires electrocytes to depend on post-transcriptional events to maintain an incomplete muscle phenotype.

Little is known about the direct contribution of post-transcriptional events in the maintenance and plasticity of muscle genes after cell differentiation is complete. Few studies have shown that the 3'-untranslated regions (3' UTR) of some mRNAs that encode sarcomeric MHCs and synaptic proteins in muscle cells are targeted for translational regulation (*Kiri & Goldspink, 2002*; *Vracar-Grabar & Russell, 2004*; *Deschênes-Furry et al., 2005*). Noncoding RNAs that interact with 3' UTRs, specifically miRNAs, are emerging in recent years as a new layer of regulators of gene expression. In fact, some miRNAs have been characterized to be skeletal-muscle specific based on their effects on proliferation, differentiation, and determination of different muscle fiber types, as well as on muscle mass homeostasis in aging muscle (*Chen et al., 2006*; *Rao et al., 2006*; *Flynt et al., 2007*; *Zacharewicz et al., 2014*; *Guess et al., 2015*; *Shi et al., 2015*). The wide phylogenetic distribution of miRNAs across the animal taxa and their conserved functional translational repression of target mRNAs through binding to 3' UTR sequences (*Ambros, 2004*; *Bartel & Chen, 2004*) suggests to us that miRNA-mediated post-transcriptional regulation may also be pivotal in the development and maintenance of the EO tissue in *S. macrurus* (Figs. 6 and 7A). Concurrent with this idea is the detection of similar amounts of fast tropomyosin (*tpma*) mRNA levels in both muscle and EO tissues (Fig. 4A) accompanied with a significantly lower detection of *tpma* protein in mature electrocytes compared to muscle fibers using immunolabeling and Western blots (R Güth & GA Unguez, 2016, unpublished data). Interestingly, the *S. macrurus* 3' UTR of *tpma* showed little similarity to that of *D. rerio* with only one predicted miRNA target site in common to both teleosts according to TargetScan Fish version 6.2 (*Lewis, Burge & Bartel, 2005*; *Ulitsky et al., 2012*) (Fig. 7B). Uncovering the functional roles of miRNAs that are upregulated (and downregulated) in the EO will help our understanding of how miRNAs work as part of a molecular toolkit involved in the modification of the muscle program to give rise to a noncontractile electrogenic tissue as the EO.

It is possible that *S. macrurus* may have evolved a unique group of miRNAs that control the expression of sarcomeric genes in EO. To this end, the existence of EO specific miRNAs was recently confirmed in a closely related gymnotiform, *Electrophorus electricus* (*Traeger et al., 2015*). Future studies will investigate whether or not *S. macrurus* contains tissue-specific miRNAs that target species-specific regulatory elements leading to the differential protein expression patterns between muscle and EO in *S. macrurus*.

Elucidation of post-transcriptional mechanisms in the formation and maintenance of electrocytes will not only have an impact on our understanding of events underlying the plasticity of skeletal muscle cells into non-contractile cells in *S. macrurus* but may also contribute to furthering our understanding of the regulation of the muscle program in

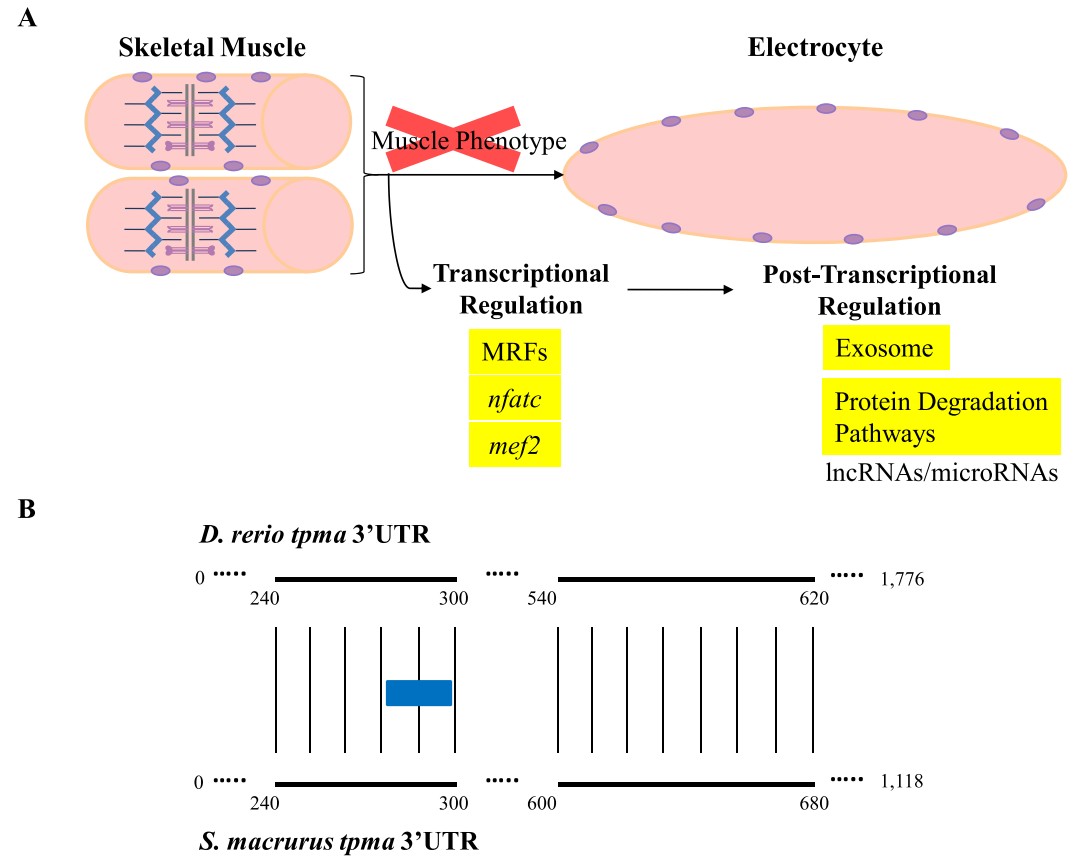

**Figure 7 Overview of expression of transcripts associated with regulation of muscle gene expression in muscle and EO of *S. macrurus*.** (A) Skeletal muscle cells are elongated, multi-nucleated, and contain structured sarcomeres. Electrocytes are large, cigar-shaped, multi-nucleated and do not contain sarcomeres, but express almost all transcripts that code for sarcomeric proteins at similar levels as muscle. In addition, the expression of muscle-specific transcription factors and protein-degradation genes is similar in both tissues, indicating that the down-regulation of the muscle phenotype in EO is not predominantly controlled at the transcriptional level or by protein degradation. These data suggest that post-transcriptional regulation of gene expression by non-coding RNAs (long non-coding RNAs, microRNAs) may play an important role in the repression of muscle gene expression in the EO of *S. macrurus*. (B) Sequence alignment of the 3′ UTRs of a fast tropomyosin transcript (*tpma*; see Fig. 3 for transcript expression ratios) from *S. macrurus* and *D. rerio* reveal very little conservation (conserved areas indicated by bold lines with narrow vertical lines between them), only one predicted miRNA target site for miRs-129/722 (indicated as a blue box) was identified in the conserved regions, indicating the possibility of many species-specific regulatory elements that may control the differential protein expression pattern between muscle and EO in *S. macrurus*. Future studies in *S. macrurus* will explore the role played by miRNAs in regulating the partial muscle phenotype of mature electrocytes. *D. rerio tpma* 3′ UTR sequence and predicted miRNA target were accessed from TargetScan Fish v. 6.2 (*Lewis, Burge & Bartel, 2005*; *Ulitsky et al., 2012*).

other cell types in vertebrates. For example, muscle transcription factors are found in almost all types of neoplastic rhabdomyosarcoma cells, which maintain a very limited myogenic phenotype (*Tonin et al., 1991*; *Wijnaendts et al., 1994*; *Frascella & Rosolen, 1998*). Similarly, muscle transcription factors expression is associated with an incomplete muscle phenotype in fully differentiated cells like Purkinje fibers of the cardiac conductive system

(*Takebayashi-Suzuki et al., 2001*), myoid cells of the thymus (*Grounds, Garrett & Beilharz, 1992*; *Kornstein, Asher & Fuchs, 1995*), and myofibroblasts from liver, kidney, and lung tissues (*Walker, Guerrero & Leinwand, 2001*).

Although our data indicate that a high level of transcriptional repression of the myogenic program is not a requisite for the emergence and maintenance of the EO phenotype in *S. macrurus*, this molecular strategy appears to diverge from that used by other electric fish species in forming electrocytes from striated muscle cells. A recent study by *Gallant et al. (2014)* found that unlike *S. macrurus*, two other South American gymnotiforms (*E. electricus* and *Eigenmannia virescens*) and one African mormyrid (*Brienomyrus brachyistius*) strongly repressed the myogenic program at the transcriptional level. A deeper analysis of the expression of transcriptionally upregulated genes in muscle compared to all three EOs of *E. electricus* using cluster analysis (*Gallant et al., 2014*) identified isomers of many sarcomeric genes. When compared to the expression of their homologs in *S. macrurus*, virtually all were similarly expressed between muscle and EO in *S. macrurus*. This adds more evidence to support our observation that *S. macrurus* utilizes a different mechanism to regulate contractile gene expression compared to other related electric fish species. A similar study by Lamanna and colleagues (*2015*) also reported a notable downregulation of structural genes associated with the sarcomere complex and MTFs known to activate them in the mature EOs of two additional mormyrids (*Campylomormyrus compressirostris* and *Campylomormyrus tshokwe*). These observations are intriguing in view of the evolutionary relationships between these electric fish species. Despite sharing a common gymnotiform ancestry, the contractile muscle gene expression profiles in EOs of *E. electricus* and *S. macrurus* differ considerably. In contrast, the EO of the gymnotiform *E. electricus* and the distantly related mormyrids *B. brachyistius*, *C. compressirostris* and *C. tshokwe* exhibit a more similar downregulation of genes associated with the contractile muscle phenotype. Given their similar transcription-dependent downregulation of MTFs and their target muscle genes, a model of convergent evolution of EOs may be at work in electric fish species with the exception of *S. macrurus* (*Gallant et al., 2014*). Moreover, that mature electrocytes in *S. macrurus* represent myogenically derived cells that became highly specialized non-contractile electrogenic cells without changing their striated muscle transcriptome provides new scenarios by which to elucidate how transcriptional and post-transcriptional events may regulate specific contractile properties in a vertebrate.

## CONCLUSIONS

The present data showed that while mature electrocytes differentiate into highly specialized electrogenic cells with a morphology and function that significantly contrast those of their muscle precursors (*Unguez & Zakon, 1998*), they do so with relatively little modification of their myogenic transcriptome. Ultrastructural and biochemical studies have failed to detect many of the sarcomeric proteins in electrocytes (*Unguez & Zakon, 1998*; *Güth, Pinch & Unguez, 2013*). However, our expression analysis showed that electrocytes continue to transcribe genes responsible for the force generation capacity of striated muscle including those that encode proteins of all sarcomeric regions, the excitation-contraction coupling

system, and the transcriptional regulators and degradation pathways known to regulate the manifestation of many of these muscle-specific genes. These data underscore the complexity of the multi-level molecular processes that must exist to generate and maintain the fully functional skeletal muscle phenotype. What is more, these data provide compelling evidence corroborating an electrocyte differentiation process in which downregulation of the muscle program takes place downstream of the transcription process. Specifically, our findings support a model of gene expression regulation in which select skeletal muscle modules or gene subsets can be adapted to give rise to a current-producing cell that no longer contracts. Whether the mismatch of transcript and protein expression of muscle genes in EO is dependent on processes that regulate mRNA transport out of the nucleus, mRNA processing, mRNA sorting, mRNA repression, or mRNA decay (*Von Roretz et al., 2011*; *Neguembor, Jothi & Gabellini, 2014*) is an exciting next step to investigate. This comprehensive analysis of parallel mRNA and miRNA profiles can inform studies on the evolution of other muscle-derived non-contractile cells such as heater organs (*Block, 1986*) and bioluminescent tissues (*Johnston & Herring, 1985*) in other teleosts, and the various muscle-like cells in vertebrates including myofibroblasts and Purkinje fibers of the cardiac conduction system.

## ACKNOWLEDGEMENTS

The authors are very indebted to Michael McDowell for the *in situ* data, Heather Sneed for assistance with analysis of gene expression data, and Chiann-Ling C Yeh for artistic work of figures for sarcomere.

### Funding

The following grants funded this work: NIH SCORE Grant 1SC1GM092297-01A1 (GAU), NSF Grant CNS-1248109 (GAU), and Howard Hughes Medical Institute Science Education Grant 5200693. The funders had no role in study design, data collection and analysis, decision to publish, or preparation of the manuscript.

### Grant Disclosures

The following grant information was disclosed by the authors:
NIH SCORE: 1SC1GM092297-01A1.
NSF: CNS-1248109.
Howard Hughes Medical Institute Science Education: 5200693.

### Competing Interests

The authors declare there are no competing interests.

### Author Contributions

- Matthew Pinch performed the experiments, analyzed the data, wrote the paper, prepared figures and/or tables, reviewed drafts of the paper.

- Robert Güth and Alexander Chaidez performed the experiments, analyzed the data, prepared figures and/or tables, reviewed drafts of the paper.
- Manoj P. Samanta conceived and designed the experiments, performed the experiments, analyzed the data, contributed reagents/materials/analysis tools, wrote the paper, prepared figures and/or tables, reviewed drafts of the paper.
- Graciela A. Unguez conceived and designed the experiments, analyzed the data, contributed reagents/materials/analysis tools, wrote the paper, reviewed drafts of the paper.

## Animal Ethics

The following information was supplied relating to ethical approvals (i.e., approving body and any reference numbers):

New Mexico State University Institutional Animal Care and Use Committee, 2014-044.

## DNA Deposition

The following information was supplied regarding the deposition of DNA sequences:

All Illumina Hi-Seq data described in this study have been deposited in the NCBI BioProject Database (accession: PRJNA248545).

## Supplemental Information

Supplemental information for this article can be found online at http://dx.doi.org/10.7717/peerj.1828#supplemental-information.

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
