# Peer review of "The myogenic electric organ of Sternopygus macrurus: a non-contractile tissue with a skeletal muscle transcriptome"

_PeerJ, doi:10.7717/peerj.1828_

## Round 0.1 · original submission · Major Revisions

Your manuscript has been reviewed by three expert reviewers. All three reviewers found the aim and approach of your work very interesting and important, but two of the reviewers raised substantial criticisms related to the methods, statistical analysis, validity of the findings of the manuscript and the interpretation of the data.

Therefore, additional experiments and major revision of the manuscript are necessary to justify the conclusions. I agree that the the main weakness of the manuscript is that a single sample of muscle and electric organ was used for transcriptome sequencing. Increasing the sample number would lead to more representative and more convincing data. The validation experiments should be more convincing and the signal/noise ratio should be more carefully adjusted. A more detailed description of the bioinformatic, statistical and other methods is needed and the comparison of the results with existing literature data is also recommended.

I suggest considering all the thoughtful and important comments of the reviewers and revising your manuscript in the light of their criticisms. Please provide your point-by-point response to the reviewers’ comments, when submitting your revised manuscript. Please explain clearly, if you do not agree with any of their comments or suggestions.

I look forward to receiving your revised manuscript.

Reviewer 1 ·

Basic reporting

The writing style of the manuscript is excellent.

Experimental design

The experimental design is excellent.

Validity of the findings

There are two fundamental problems with the data as currently reported. First, as shown by the validation experiments in Table 3, as well as the scatter plot in Figure 1, there seems to be considerable 'noise' (false positives and potentially false negatives) in the data selected for interpretation. The authors give relatively little information on any statistical thresholds utilized in data analyses - they say, "removal of sequences with fewer than 100 copies in both EO and skeletal muscle to minimize inclusion of genes likely to be expressed in cell types other muscle cells and electrocytes." This reviewer has trouble making sense of this statement. If a transcript shows 1,000,000 RNA reads in electric organ, and 0 copies in skeletal muscle, then this is discarded from the analysis? This analysis would certainly lead to similar expression patterns between electric organ and muscle, because that is precisely what was selected for. This is akin to 'throwing out the baby with the bathwater'. Further evidence of challenges in signal/noise balance is the validation experiments where RNAseq data was compared to Nanostring and RT-PCR. There is very little concordance of data by the three methods. Due to these concerns, much of the remaining data seems over-interpreted, as many of the figures may deal with noise rather than true differences. The authors must be much more descriptive and robust regarding their bioinformatic and statistical methods before launching into the many conclusions they now derive.

The second major issue is a lack of compare/contrast to the existing literature. There have been other genome-wide studies of electric organs, yet none are compare/contrast to the current data, and two of the most relevant are not even cited. The Discussion must do a careful compare/contrast with each of the papers below.

Gallant JR, Traeger LL, Volkening JD, Moffett H, Chen PH, Novina CD, Phillips
GN Jr, Anand R, Wells GB, Pinch M, Güth R, Unguez GA, Albert JS, Zakon HH,
Samanta MP, Sussman MR. Nonhuman genetics. Genomic basis for the convergent
evolution of electric organs. Science. 2014 Jun 27;344(6191):1522-5. doi:
10.1126/science.1254432. PubMed PMID: 24970089.


Lamanna F, Kirschbaum F, Tiedemann R. De novo assembly and characterization of
the skeletal muscle and electric organ transcriptomes of the African weakly
electric fish Campylomormyrus compressirostris (Mormyridae, Teleostei). Mol Ecol
Resour. 2014 Nov;14(6):1222-30. doi: 10.1111/1755-0998.12260. Epub 2014 Apr 23.
PubMed PMID: 24690394.

Nazarian J, Berry DL, Sanjari S, Razvi M, Brown K, Hathout Y, Vertes A, Dadgar
S, Hoffman EP. Evolution and comparative genomics of subcellular specializations:
EST sequencing of Torpedo electric organ. Mar Genomics. 2011 Mar;4(1):33-40. doi:
10.1016/j.margen.2010.12.004. Epub 2011 Feb 12. PubMed PMID: 21429463; PubMed
Central PMCID: PMC3412124.

Additional comments

The abstract states that the aim of the study was “To examine the molecular regulatory pathways that control the myogenic program in electrocytes, we have sequenced EO and skeletal muscle transcriptomes from S. macrurus.” But from the abstract it is not clear if this was achieved or not (the abstract is quite descriptive/generic). The authors need to give some specifics regarding their findings relative to their intent. To be more specific, “complexity of the multi-level molecular processes” – what multi-level processes and what complexities? Indeed, there is nothing in the abstract that gives a sense of the results of the study.

Reviewer 2 ·

Basic reporting

This is an exciting MS that really puts the “Principle of Krogh” into action by choosing a model that is well suited and amenable to the objectives of their study. The question “the way in which MTFs modulate (or maintain) the muscle phenotype after cell differentiation” is of broad scientific interest as the general character of the question is not restricted to muscle and thus the MS provides insight relevant for a number of other tissue and research fields. The approach is well described and ingenious establishing a muscle transcriptome for the electric organ (derived from striated muscle/ express MTF/ down regulate myogenic factors) and comparing it with the skeletal muscle transcriptome. The Introduction is clear and to the point and the rational for studying S. macrurus is well presented. As might be predicted the results are complex and do not give a clear cut answer but they may provide the basis for future studies directed at understanding the evolution of muscle derived non-contractile cells. The approach is that of a standard transcriptome study, what makes the study of high interest is the way the approach has been used to answer a very specific question. Unfortunately there are a number of negative aspects in relation to this MS, that curtailed my initial enthusiasm. The results and discussion could be more focused on the research question and the authors have become rather “swamped” by their data. However, the main limitation and weakness of this MS is the strategy taken for sequencing, a single sample of muscle and EO. The authors use RNA-seq and suggest that the Nanostring analysis, qPCR and in situ analysis validate their sequencing but the other approaches are not sufficiently comprehensive to robustly validate the transcriptome data in my opinion.

Experimental design

1. My main reservations about the MS are associated with the methods. The actual assembly methods and approach appear satisfactory. However, sample number for transcriptome sequencing is a problem. I understand that the cost of sequencing can be prohibitive for large scale studies but the minimum requirement for transcriptomes in my opinion should be at least 2 samples / tissue so it is possible to verify that the results are minimally representative and not atypical (or a one-off) since conservation of at least the main characteristics of the transcriptome between individuals can be confirmed. Alternatively by using several individuals in a single sequencing exercise at least a “representative” transcriptome is obtained. For this reason I have reservations about how representative the transcriptome results are and how valid it is to use them to define differential transcripts. Moreover, the analysis of 60/22 transcripts does not really resolve this problem as they are a tiny part of the total transcriptome.

2. I have identified a number of other methodological issues that also need to be resolved primarily linked with lack of information or the way the methods have been written that makes them difficult to follow. Furthermore the rational for certain approaches eg. slection of genes for further analysis is unclear. Some examples now follow.
pp6, ln 1: please provide a complete description of fish husbandry conditions as this is essential to establish if fish were under stress, growing or losing muscle mass. How long were the fish housed in the institution before they were used for sampling? Were the fish sampled for muscle and EO killed or allowed to survive it is currently unclear from the description.
Pg 7, ln 4 the authors indicate that further reads from the EO of spinally transected fish were included in the general transcriptome assembly of EO and muscle. Since this data is unpublished the authors should at least provide some description of the quality/characteristics of the data – was it processed in the same way as the EO and muscle that is reported in this study?
Pg 9, ln1-7: calculation of gene expression level; I am not convinced by the EO vs muscle comparison and subsequent identification of differential transcripts. Statistically this analysis does not hold up since it is on a single sample of each tissue. Have you verified if the counts of rps11, cct5, and snrpb, the reference genes for qPCR vary between the 2 libraries?
Pg 9, ln 11: I do not understand why only 22 of the 60 transcripts tested are reported in this MS. In relation to the total number of assembled contigs obtained 22 is a tiny fraction and does not robustly confirm the results of the sequencing of 1 individual. Did the authors include the sample used for transcriptome sequencing in their NanoString quantification analysis and quantitative PCR? Table 2 that indicates expression levels of muscle specific transcripts is unclear (no indication of what is represented in the columns is given – is it count number normalized for total reads). No rational is provided for the selection of genes used for validation of the transcriptome.
Pg 12, ln 1: shorten and summarize this section about selection of reference genes and modify the section pg 9, ln 16: about reference genes it is confusing.
The rational for carrying out in situ and the 2 genes chosen is unclear. If it is to shown muscle then histology would be sufficient. If it was to validate the sequencing results then more transcripts should have been mapped.
The in situ figure is not of sufficient quality to really verify what is stained. I suggest the autors provide a H&E image so tissue disposition can be seen and then present their in situ results preferably in colour so it is possible to distinguish the localization of the AP stain and distinguish it from melanin. The designation of + and – is not sufficiently informative. Do the images represent transverse sections? The authors should aim to include some further annotation highlighting key features in the section. The scale bars are missing from some of the images.

Validity of the findings

Pg 19, ln 20: I do not agree with the authors’ statement, “This multi-platform validation approach gives us high confidence that the RNA-seq data we have collected is an accurate representation of transcript levels in EO and muscle of S. macrurus.”
The authors have assembled >100,000 contigs, they select and analyse 60 by NanoString quantification, then 22 by qPCR and only 2 transcripts (myl and myh) by in situ. I do not consider that a robust validation of the RNA-seq data from 1 individual is performed and so I am convinced that the differential transcriptome the authors report from EO and muscle are valid or are representative of this tissue from multiple individuals.

·

Basic reporting

No comments

Experimental design

No comments

Validity of the findings

No comments

Additional comments

I was thoroughly impressed by this paper. The study is extremely well designed and executed. The introduction contains a balanced review of the subject and an excellent justification for the study. The methods and results are very carefully reported and discussed. The findings of this study are intriguing and important, and will certainly stimulate new directions of research. Particularly appealing aspects of the manuscript are that it is accessible to a non-specialist (such as this reviewer) in gene expression studies/transcriptome analyses and muscle/electric organ biology, and that it provides an evolutionary context. The manuscript is very well written and requires in my opinion almost no modification. I found just a few trivial issues, which can easily be corrected. These are listed below.
Abstract L5. Misspelling of Sternopygus
P8.L.7. Spell out Electrophorus and Danio in full on first mention.
24. Change Gymnotiform to gymnotiform (only Gymnotiformes should have a capital G)
Fig. 1 appears pixelated

---

## Round 0.2 · accepted · Accept

The Reviewers are satisfied with the scientific and technical improvement of the revised manuscript. I have to remind you, however, that Reviewer 2 found several spacing problems, misspellings and other typing errors in the revised manuscript. Spacing problems occur mostly in the Materials and Methods (e.g 100bp instead of 100 bp, 0.1M instead of 0.1 M, etc), but there are some inconsistencies throughout the manuscript (e.g. 3'-UTR, 3'UTR and 3' UTR). As the Journal does not offer editing services, it is the responsibility of the authors to provide an article which is “written in English using clear and unambiguous text and must conform to professional standards of courtesy and expression”. Therefore, I strongly advise you to check very carefully the manuscript for typos, misspellings, inconsistencies and other language problems and correct those at the proof stage.

Reviewer 2 ·

Basic reporting

The authors have modified the approach and use the Next Gen sequencing to inform their qPCR analysis. This overcomes the problem with regards to inference made on the basis of analyzing a single individual as now multiple individuals are targeted for the candidate genes analyzed. Technical issues related to the difficulty of discriminating the in situ signal with Akl Phos and the melanin is overcome by using digoxigenin and an alternative chromagen. The modified approach taken by the authors has made the results more robust and convincing.
Apart from a plethora of typos and misspelling this MS is suitable for publication.

Experimental design

See comments to previous submission. The revision to the methods has made the experimental approach taken by the authors easier to understand.

Validity of the findings

The shift in the way the authors analyze their data has overcome the issue highlighted in my previous review. The authors have made their results and discussion more focused and it is now less speculative and result findings reported supported by more robust analysis.

Additional comments

By revising the way in which the authors use the Next Gen sequencing approach (to inform the PCR analysis), by increasing the number of qPCR analysis focused on candidate genes associated with the question they are addressing the authors have resolved the majority of my concerns. The methods are also now clearer and better explain the experimental approach. The discussion is more focused and appropriately cites the literature
I am suggesting minor revisions because the authors need to carefully proof their MS and remove a lot of small errors (eg. lack of spaces, misspelling etc).
I have not gone through the MS and listed all the changes as I think the onus is on the author to deliver an MS without errors.

·

Basic reporting

No comments

Experimental design

No comments

Validity of the findings

No comments

Additional comments

The reviewers have carefully addressed all of the concerns raised by the reviewers, and the manuscript (as before), is very clear and well written. Despite the technical complexity of the paper, the emergent results, and their unusual nature in the light of the findings published recently for other gymnotiform (and mormyriform) electric fish are very well described. My role in reviewing this paper is limited because I have no expertise in transcriptomics or NGS. Nonetheless, based on my knowledge of electric fish biology I saw no flaws at all in the conceptual approaches, methods, and in the presentation of the findings. The identification of the unusual regulation of contractile gene expression in Sternopygus relative to other gymnotiforms (which appear to show convergent similarities in this regard with mormyriforms, along with multiple other aspects of the electrogenic and electrosensory system) is an important discovery and will springboard many additional studies. In conclusion, this is an important contribution to our understanding of electrogenesis and the extent to which the myogenic program is repressed at the level of transcription.